# Towards a Unified View on Visual Parameter-Efficient Transfer Learning

## Abstract

Since the release of various large-scale natural language processing (NLP) pre-trained models, parameter efficient transfer learning (PETL) has become a popular paradigm capable of achieving impressive performance on various downstream tasks. PETL aims at making good use of the representation knowledge in the pre-trained large models by fine-tuning a small number of parameters. Recently, it has also attracted increasing attention to developing various PETL techniques for vision tasks. Popular PETL techniques such as Prompt-tuning and Adapter have been proposed for high-level visual downstream tasks such as image classification and video recognition. However, Prefix-tuning remains under-explored for vision tasks. In this work, we intend to adapt large video-based models to downstream tasks with a good parameter-accuracy trade-off. Towards this goal, we propose a framework with a unified view of PETL called visual-PETL (V-PETL) to investigate the effects of different PETL techniques, data scales of downstream domains, positions of trainable parameters, and other aspects affecting the trade-off. Specifically, we analyze the positional importance of trainable parameters and differences between NLP and vision tasks in terms of data structures and pre-training mechanisms while implementing various PETL techniques, especially for the under-explored prefix-tuning technique. Based on a comprehensive under-standing of differences between NLP and video data, we propose a new variation of prefix-tuning module called parallel attention (PATT) for video-based downstream tasks. An extensive empirical analysis on two video datasets via different frozen backbones has been carried and the findings show that the proposed PATT can effectively contribute to other PETL techniques. An effective scheme Swin-BAPAT derived from the proposed V-PETL framework achieves significantly better performance than the state-of-the-art AdaptFormer-Swin with slightly more parameters and outperforms full-tuning with far less parameters.

## 1 Introduction

Many vision tasks rely on fine-tuning pre-trained models to achieve good performance. One standard modus operandi of transfer learning consists of two steps: pre-train a model on a source domain and fine-tune the entire model on a target domain (Zhuang et al., 2020). Despite that prior works have achieved promising performance, such vanilla practice of fine-tuning is faced with challenges for adopting large models to downstream tasks. This full-tuning strategy requires one to update and store separate model parameters for different downstream tasks, which can be expensive and infeasible for the era of increasingly large models from EfficientNet-based (Pham et al., 2021) (480M parameters) to Transformer-based (Yu et al., 2022) (2, 100M parameters) ones. For such large models, making good use of shared parameter weights deployed on the cloud can be beneficial for edge devices such as autonomous vehicles, drones who are intensive in computing and battery resources (Yuan et al., 2022). Second, the full fine-tuning strategy relies on high-quality downstream data and can hardly adapt to unseen scenarios that have large distribution shift (Kumar et al., 2021), which is unlike the learning process of humans who can learn from few samples and generalize well to new circumstances. This issue has been researched in directions such as zero-shot learning, few-shot learning, and continual learning (Li et al., 2021a). Another popular strategy is fine-tuning the downstream task head, i.e., the last fully connected (FC) layer, to avoid tuning the whole backbone model, which usually leads to poor performance when the target domain is large in data scale (see Figure

1). Given the paradigm of fine-tuning increasingly large models, how to transfer such large models with parameter-accuracy trade-off is a hot topic in various domains (Gusak et al., 2022; Sung et al., 2022; Lin et al., 2020; Houlsby et al., 2019).

Taking the video-based action recognition task as an example, it can be inconvenient for deploying such large models to edge devices such as an autonomous driving (Liu et al., 2019) and unmanned aerial vehicle (Li et al., 2021b) as they can heavily rely on the interaction with cloud services for adapting to new environments via active learning (Wang et al., 2021) or continual learning (Li et al., 2021a). Re-training large models on the cloud are usually not cost-effective due to the expensive overheads of storage and computational resources. Furthermore, these resources are limited on edge devices such as autonomous vehicles and unmanned aerial vehicles, making the sense for developing effective fine-tuning methods with proper parameter-accuracy trade-off that can be fine-tuned on edge devices and interacting with the large models deployed on the cloud.

There have been some pioneering works for the PETL of visual models such as AdaptFormer (Chen et al., 2022) and visual prompt tuning (VPT) (Jia et al., 2022). AdaptFormer is primarily proposed based on vision transformer (Zhai et al., 2022), representing one of the state-of-the-art large models for image-based tasks. The proposed adapter module directly brings from Houlsby et al. (2019) due to its convenience of being inserted to any models. Implementing with a large batch size of $1,024$ with $64$ GPUs, Adaptformer shows promising parameter-accuracy trade-off on video data. However, such powerful computing resource is not realistic for the usage of edge devices. Meanwhile, whether the good trade-off can be maintained for small batch size remains under-explored. Inspired by the Prompting in NLP (Liu et al., 2021), VPT proposes visual-prompt to fine-tune visual models for image-based tasks. According to the empirical results in Chen et al. (2022), adapter modules achieves superior performance over VPT in the regimes of both self-supervised and supervised

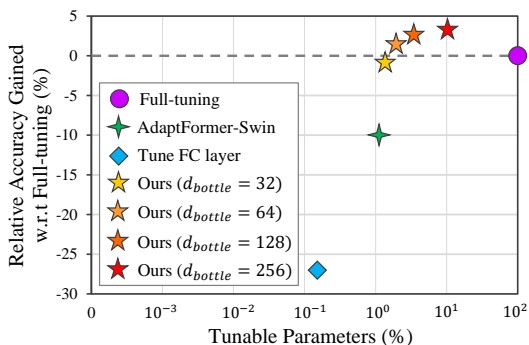

Figure 1: Parameter-accuracy trade-off. Adapting backbone Swin-B (Liu et al., 2022) pre-trained on Kinetics 400 via different fine-tuning methods on the something-something v2 (Goyal et al., 2017) dataset. Our methods perform significantly better than the state-of-the-art AdaptFormer-Swin (Chen et al., 2022) (our implementation with batch size 16) with slightly more tunable parameters, and outperform full-tuning with increasing margins when using larger values of $d_{bottle}$.

pre-training. Another concern of VPT is its modification to the original model parameters might affect the knowledge representation of backbone models. Hence, we do not continue to compare our method with VPT but comparing with the adapter on video-based downstream tasks.

Taking the recent inspiration of the mix-and-match adapter (MAM adapter) (He et al., 2022a) in NLP, we aim to propose a unified model for the vision domain, especially for video-based downstream tasks. He et al. (2022a) analyzed the unified view among PETL techniques such as prefix-tuning, low-rank (LoRA) adaptation, and adapter, pointing out the similarity between prefix-tuning and adapter in terms of calculating the attention. The difference is that the former performs weighted addition while the latter ones is unweighted. Note that prefix-tuning has not ever been applied to visual tasks in the form of pure visual models due to the intrinsic differences regarding pre-training methods of NLP and vision models. Another obstacle of directly applying prefix-tuning to visual tasks is the structural difference between text and vision data (we further discuss this in Section 2.3). Considering the video-based action recognition task, we propose a new variation of the prefix-tuning module called parallel attention (PATT) to adapt video-based pre-trained large models to downstream domains with varied data scales. The differences of our method comparing the original prefix-tuning in NLP are twofold: prefix calculation and the manner of insertion (see Figure 2[b] and Figure 3). Regarding the backbone model, we focus on Video Swin Transformer (Liu et al., 2022), one of the state-of-the-art vision models that bring competitive performance on large-scale action recognition datasets such as Kinetics 400 and 600 Kay et al. (2017).

Our main contributions can be threefold as follows:
1. We analyze different PETL techniques using the backbone model Swin Video Transformer for

video-based tasks, providing a unified view via our V-PETL framework and investigating the importance of the fine-tuning position.

2. Based on the comprehensive understanding of intrinsic differences between NLP and video data regarding data structures and pre-training mechanisms, we leverage prefix-tuning to our V-PETL with a new variation called PATT.

3. Upon extensive ablation experiments regarding various effect factors, we empirically validate the promising parameter-accuracy trade-off achieved by our adjustable and easy-to-use PATT module, contributing to the existing literature of PETL techniques.

## 2 UNIFIED FRAMEWORK

### 2.1 RECAP OF VIDEO SWIN TRANSFORMER

Video Swin Transformer (Liu et al., 2022) is formed with Transformer layers (a.k.a. stages) that are consisted with 3D Video Swin Transformer blocks. With varied layers, blocks, and channel sizes, the model can be formed as Swin-T, Swin-S, Swin-B, and Swin-L. The basic architecture of a 3D Swin Transformer block is shown in Figure 2, which is mainly composed of a 3D shifted window-based multi-head self-attention (3DSW-MSA) module and a fully connected feed-forward network (FFN) implemented with a 2-layer MLP. Layer normalization (LN) and residual connection are respectively performed before and after both FFN and 3DSW-MSA modules. One such Video Swin Transformer block can be represented as:

$$\begin{aligned} \hat{\boldsymbol{Z}}^l &= \text{3DSW-MSA}(\text{LN}(\boldsymbol{Z}^{l-1})) + \boldsymbol{Z}^{l-1}, \\ \boldsymbol{Z}^l &= \text{FFN}(\text{LN}(\hat{\boldsymbol{Z}}^l)) + \hat{\boldsymbol{Z}}^l, \end{aligned} \tag{1}$$

where $\hat{\boldsymbol{Z}}^l$ and $\boldsymbol{Z}^l$ respectively indicate the output of 3DSW-MSA and FNN modules.

Given a video input sized $t \times w \times h \times 3$, containing $t$ video frames with their heights and widths being $h$ and $w$, respectively. The 3D patch for video data sized $2 \times 4 \times 4 \times 3$ is treated as a token. Then we will have $\frac{t}{2} \times \frac{w}{4} \times \frac{h}{4}$ 3D tokens after a 3D patch partitioning layer. Given the 3D tokens sized $\frac{t}{2} \times \frac{w}{4} \times \frac{h}{4}$ and a 3D window with the size of $p \times m \times m$, the self-attention module, using the regular window partition strategy, will partition the 3D tokens to $\frac{t}{2p} \times \frac{w}{4m} \times \frac{h}{4m}$ non-overlapping windows. For shifted 3D window, the partition is shifted along the temporal, height, and width dimensions by $\frac{p}{2} \times \frac{m}{2} \times \frac{m}{2}$. For example, if we have an input video sized $8 \times 224 \times 224 \times 3$ and a $8 \times 7 \times 7$ 3D window, after the patch embedding, we will have $4 \times 56 \times 56$ 3D tokens with each of them sized $2 \times 4 \times 4 \times 3$. Without shifting, the non-overlapping window size will be $1 \times 8 \times 8 = 64$. Then through the 3D window shifted by $(4, 3, 3)$, the number of 3D windows becomes $1 \times 9 \times 9 = 81$.

The 3DSW-MSA module is formed with a 3D relative position bias $\mathbf{B} \in \mathbb{R}^{p^2 \times m^2 \times m^2}$, each of which can be represented as:

$$Attention(\mathbf{Q}, \mathbf{K}, \mathbf{V}) = SoftMax(\frac{\mathbf{Q}\mathbf{K}^T}{\sqrt{d}} + \mathbf{B})\mathbf{V}, \tag{2}$$

where $\mathbf{Q}, \mathbf{K}, \mathbf{V} \in \mathbb{R}^{p \times m \times m \times d}$ are the query, key, and value matrices, $p \times m \times m$ is the number of tokens and $d$ is the dimension of the tokens. MSA simultaneously performs the attention mechanism for $n_{head}$ heads, where the $i$th head can be parameterized by $\boldsymbol{W}_q^{(i)}, \boldsymbol{W}_k^{(i)}, \boldsymbol{W}_v^{(i)} \in \mathbb{R}^{d \times 3d}$, projecting the input $Z^{l-1}$ to queries, keys, and values. Given a matrix $\boldsymbol{C} \in \mathbb{R}^{\widetilde{m} \times d}$, $\widetilde{m} = p \times m \times m$, for performing attention, the 3DSW-MSA can be calculated as:

$$\begin{aligned} \text{3DSW-MSA}(\boldsymbol{Z}^{l-1}, \boldsymbol{C}) &= Concat(head_1, ..., head_n)\boldsymbol{W}_o, \\ head_i &= Attention(\boldsymbol{Z}^{l-1}\boldsymbol{W}_q^{(i)}, \boldsymbol{C}\boldsymbol{W}_k^{(i)}, \boldsymbol{C}\boldsymbol{W}_v^{(i)}), \end{aligned} \tag{3}$$

where $\boldsymbol{W}_o$ is the parameters of a linear project layer. The FNN module is composed of two linear layers with a GELU activation function in between, which can be computed as:

$$\text{FFN}(\hat{\boldsymbol{Z}}^l) = \text{GELU}(\text{LN}(\hat{\boldsymbol{Z}}^l)\boldsymbol{W}_1 + \boldsymbol{b}_1)\boldsymbol{W}_2 + \boldsymbol{b}_2, \tag{4}$$

where $\boldsymbol{W}_1 \in \mathbb{R}^{d_{hidden} \times d}$, $\boldsymbol{W}_2 \in \mathbb{R}^{d \times d_{hidden}}$, $\boldsymbol{b}_1 \in \mathbb{R}^{d_{hidden}}$, and $\boldsymbol{b}_2 \in \mathbb{R}^d$. The value of $d_{hidden}$ usually takes a large value (e.g., $d_{hidden} = 4d$).

## 2.2 RECAP OF PETL TECHNIQUES

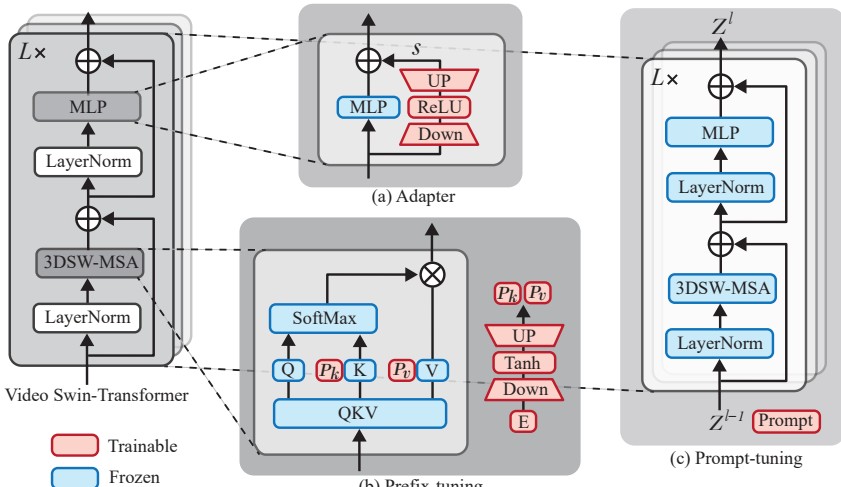

Figure 2: V-PETL: A unified view of visual PETL techniques. They bring trainable parameters to different positions of the backbone model with various manners. AdaptFormer and Prefix-tuning respectively perform at the MLP and 3DSW-MSA modules that can adjust the number of trainable parameters via the bottleneck size of down and up projections. While prompt-tuning performed at the layer-level can adjust the length of prompts to control the tuned parameters.

**Prefix-tuning (Li & Liang, 2021)**: The prefix-tuning approach prepends learnable prefix tokens to the keys and values of the MSA module of the model (see Figure 2[b]). Specifically, two prefix matrices $\boldsymbol{P}_k, \boldsymbol{P}_v \in \mathbb{R}^{d_{token} \times d}$ that are randomly initialized with $d_{token}$ tokens and transformed from two linear layers (with parameters $\boldsymbol{W}_{pk}^{(i)} \in \mathbb{R}^{d \times d_{middle}}$ and $\boldsymbol{W}_{pv}^{(i)} \in \mathbb{R}^{d_{middle} \times d}$) and a Tanh layer in between are concatenated to the original key and value, leading the calculation of $head_i$ in Eq. 3 to:

$$head_i = Attention(\boldsymbol{Z}^{l-1}\boldsymbol{W}_q^{(i)}, concat(\boldsymbol{P}_k^{(i)}, \boldsymbol{C}\boldsymbol{W}_k^{(i)}), concat(\boldsymbol{P}_v^{(i)}, \boldsymbol{C}\boldsymbol{W}_v^{(i)})), \qquad (5)$$

where the $concat$ is the concatenation performed along the token dimension to mimic the prefix-tuning in NLP tasks. Here, a question regarding whether this direct implementation will work for the vision domain is raised (results are in Table 4). This direct implementation is empirically invalid and we make further modification on it in Section 2.3.

**Adapter (Chen et al., 2022)**: Inspired by the works of Houlsby et al. (2019); He et al. (2022a) for PETL in NLP tasks, adapter (Chen et al., 2022) has been directly used for vision tasks, showing promising performance using far less tunable parameters. The number of parameters of adapter is controlled by a parameter $d_{bottle}$ ($d_{bottle} \ll d$), adjusting the space size of a low-dimensional representation. The adapter module first uses a down-projection with $\boldsymbol{W}_{down} \in \mathbb{R}^{d \times d_{bottle}}$ to project the feature to the lower-dimensional representation, followed by a ReLU activation function, and a up-projection with $\boldsymbol{W}_{up} \in \mathbb{R}^{d_{bottle} \times d}$.

$$\widetilde{\boldsymbol{Z}}^l = \text{ReLU}(\text{LN}(\hat{\boldsymbol{Z}}^l)\boldsymbol{W}_{down})\boldsymbol{W}_{up}, \qquad (6)$$

then two positions implementing adapter (parallel and sequential) can be respectively computed as:

$$\boldsymbol{Z}^l = \text{FFN}(\text{LN}(\hat{\boldsymbol{Z}}^l)) + \hat{\boldsymbol{Z}}^l + s\widetilde{\boldsymbol{Z}}^l,$$
$$and \; s\boldsymbol{Z}^l = \text{ReLU}(\text{FFN}(\text{LN}(\hat{\boldsymbol{Z}}^l))\boldsymbol{W}_{down})\boldsymbol{W}_{up} + \hat{\boldsymbol{Z}}^l, \qquad (7)$$

where $s$ is a scalar, controlling the effect of the adapter (will be ablated in experiments). According to Chen et al. (2022), the parallel implementation (see Figure 2[a]) empirically performs better.

**Prompt-tuning (Jia et al., 2022)**: Prompt-tuning (see Figure 2[c]) is inspired by the success of prompt-tuning that adapts large scale models to varied downstream NLP tasks. The idea of VPT (Jia et al., 2022) is to fine-tune a learnable matrix $\boldsymbol{P}_{prompt}^{l-1} \in \mathbb{R}^{d_{prompt} \times d}$, $d_{prompt} < d_{token} - 1$ for

the $l$th Transformer layer or all Transformer layers, which are known as shallow prompt and deep prompt, respectively.

$$\hat{Z}^l = \text{3DSW-MSA}(\text{LN}([x^{l-1}, P^{l-1}_{prompt}, Z^{l-1}])) + Z^{l-1}, \tag{8}$$

where $x^{l-1} \in \mathbb{R}^d$ denotes the [CLS]'s embedding for the $l$th layer's input space, $P^{l-1}_{prompt}$ is implemented by overlapping the top $d_{prompt}$ tokens of $Z^{l-1}$ (Jia et al., 2022). While it has also been implemented in front of the $x^{l-1}$ (Chen et al., 2022).

**Others**: Other PETL techniques include ST-Adapter Pan et al. (2022), LoRA (Hu et al., 2022), and BitFit (Zaken et al., 2022). ST-Adapter mainly adapts image-text models pre-trained on large scale datasets such as 400M image-text pair proposed by CLIP (Radford et al., 2021) and the IG-3.6B used by SWAG (Singh et al., 2022) to video understanding downstream tasks, which matches and even outperforms full-tuning. LoRA approximates the optimization process by injecting learnable low-rank matrices into the attention module. This method does not show superior performance for NLP tasks in terms of parameter efficiency. Hence, we do not prioritize this direction in this work. BitFit only tunes the bias terms of the backbone models, making it very parameter-efficient.

### 2.3 REVISITING PREFIX-TUNING FOR VISUAL TASKS

The prefix implementation in NLP Li & Liang (2021); He et al. (2022a) can be regarded as prepending contextual information for downstream tasks, which is similar with the pre-training process aiming to predict masked words in the process of an inner loop (Brown et al., 2020). Considering the pre-training process of pure vision models, such direct implementation might not make sense for visual tasks. Although such autoregressive pre-training has been conducted in visual domain (He et al., 2022b; Tong et al., 2022), but adding prefix for a sentence input in NLP can be structurally different with the visual domain. Specifically, masked pixels in image or video data cannot be regarded as some word level semantic information (e.g., a subject or an action) as in the NLP.

Recall that the embedding state of prefix-tuning is randomly initiated, which is known as learnable prefix but can bring random noise that later turns out affecting the convergence of the fine-tuning downstream tasks. Hence, inspired by the connection between adapter and prefix (He et al., 2022a), we avoid such learnable prefix design with random initialization and propose a parallel attention (PATT) to the original attention module (see Figure 3). The adapter structure can effective control the number of trainable parameters via $d_{bottle}$, which is similar with the effect of the middle dimension $d_{middle}$ of $W^{(i)}_{pk}$ and $W^{(i)}_{pv}$ for preparing the prefix. Specifically, for the $l$th layer, we use output of its previous layer $Z^{l-1}$ and project it to a pair of matrices $K_p, V_p \in \mathbb{R}^{\tilde{m} \times d}$ via a similar mechanism of Eq. 6:

$$K_p, V_p = \text{Tanh}(Z^{l-1}W_{down})W_{up}, \tag{9}$$

where Tanh is the activation function used for preparing the prefix, which can be replaced by other activation functions such as RELU and GELU. Here, we follow the original prefix implementation as its value ranges from $-1$ to $1$. Given $K_p$ and $V_p$, Eq. 5 can be rewritten as:

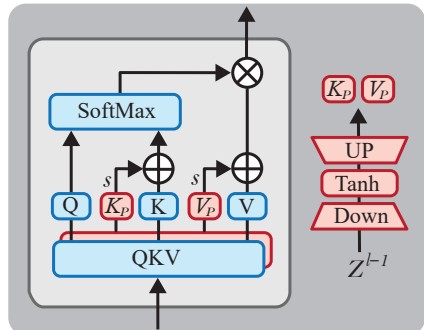

Figure 3: Structure of PATT. Red parts are trainable parameters calculated by the same input for preparing query, key, and value (i.e., the output of the previous layer passing through a layer normalization layer $Z^{l-1}$).

$$head_i = Attention(Z^{l-1}W^{(i)}_q, sK_p + CW^{(i)}_k, sV_p + CW^{(i)}_v), \tag{10}$$

where $s$ is a scalar for adjusting the effect of PATT. Note that without considering the physical meaning of such design, for PETL purpose, one can perform similar practise for any combinations of **Q**, **K**, and **V**. This brings connection to the LoRA (Hu et al., 2022) method, which add parallel trainable parameters to **Q** and **V**. Empirically, where to perform the PATT makes little difference, but the amount of trainable parameters brings larger effect for large scale downstream domains.

### 2.4 V-PETL: UNIFIED VIEW ON VISUAL PETL

Given the PETL techniques at hand, there can be many potential combinations leading to good parameter-accuracy trade-off. However, it is unrealistic to exhaustively test all the methods for a specific downstream task. Other than probing such solution via evolutionary search as in Zhang et al. (2022), we aim to propose more understandable models by empirically analyzing the effect of different designs independently. According to the preliminary results shwon in Figure 1, we argue that the position and amount of parameters are important for PETL techniques, especially when the target domain is not small.

To verify the importance of position and tuned parameter amount, we independently tune different modules of the backbone model. Table 1 shows the results. We can see that the attention module's QKV layer has 20.98M pa-

Table 1: Comparison of independently fine-tuning varied positions of the video swin transformer block on SSv2.

| Position | # Params | Top-1 (%) |
|---|---|---|
| Full-tuning | 87.82M | 50.99 |
| Tune FC Layer | 0.18M | 24.13 |
| LayerNorm 1 | 0.02M | 14.35 |
| Attn, Proj | 6.99M | 47.58 |
| Attn, QKV | 20.98M | 50.02 |
| Attn, SoftMax | 0.95M | 27.67 |
| LayerNorm 2 | 0.02M | 14.62 |
| MLP, FC1 | 27.97M | 47.10 |
| MLP, FC2 | 27.93M | 45.32 |
| DownSample | 2.76M | 27.53 |

rameters while the MLP module has the most number of parameters of 55.90M. Tuning positions with more parameters, will lead to better performance for SSv2. Thanks to the bottleneck mechanism of adapter and prefix-tuning, one can effectively achieve a good parameter-accuracy trade-off. As such, we derive a model called Swin-B-adapter-PATT (Swin-BAPAT) from the V-PETL framework by using the parallel adapter and our PATT to leverage the adaption of pre-trained backbone model at the positions of attention and MLP modules, respectively. In addition to adapter and PATT, we also fine-tune the last fully connected layer as it has relatively smaller amount of tunable parameters (i.e, 0.18M) than adapter and PATT.

## 3 EXPERIMENTS

### 3.1 EXPERIMENTAL SETTINGS

**Video Datasets**: **Something-something v2** (SSv2 (Goyal et al., 2017)) It has 108,499 short videos for 174 human-object interaction categories with durations between 2 to 6 seconds. The challenge of this dataset is that it contains 23,137 distinct object names with an imbalanced distribution. The original dataset is split into train, validation, and test sets with a ratio of 8:1:1. The extended version (SSv2) of this dataset is consisted of 168,913 training samples, 24,777 validation samples, and 27,157 testing samples with the sample number of action labels. The training and testing samples are used. **HMDB51** (Kuehne et al., 2011) contains 6,766 video samples for 51 action categories including videos of varied visible body parts, camera motion, camera view, and clip quality. All video samples have at least 101 clips and a minimum height of 60 pixels for actors. The original dataset has three splits of training and evaluation. We follow existing work Chen et al. (2022) by using the first training and evaluation split that has 3,570 and 1,530 samples, respectively. **Image Datasets**: Following the experimental set ups in AdaptFormer, three datasets CIFAIR-100 Krizhevsky et al. (2009), Street View House Numbers (SVHN) Goodfellow et al. (2013), and Food-101 Bossard et al. (2014) are used. **CIFAIR-100** has 50,000 and 10,000 training and validation images, respectively, with the resolution of $32 \times 32$ and 100 categories; **SVHN** is a digit classification dataset that has 73,257 training sample and 26,032 testing samples; **Food-101** includes 101k images of 101 food categories with each of them has 750 training and 250 testing samples.

**Implementation details**: It is worth noting that big batch size (i.e., 1,024) and the number of input video frames (i.e., 32 frames) can greatly benefit good performance (Carreira & Zisserman, 2017; Liu et al., 2022; Chen et al., 2022), which usually requires GPU clusters to enable the training. AdaptFormer (Chen et al., 2022) uses such powerful GPU cluster to achieve good performance. However, good performance might not hold when the batch size is small. Following the more common hardware device setup, we use 4 GeForce 3090 GPUs for all experiments, leading to a batch size of 64. All the experiments are fine-tuned for 70 epochs. We use the Swin-B[1] model pre-trained on Kinetics 400 and 600. For HMDB51, we report the results without tuning the FC layer due to the significant effect of the FC layer on relatively small scale dataset. Following Chen et al. (2022), we do not perform regularization strategies such as mixup, cutmix, color jittering,

---

[1] https://github.com/SwinTransformer/Video-Swin-Transformer

Table 2: Comparison of Top-1 accuracy using varied amount of parameters adjusted by $d_{bottle}$, different pre-training domains, and the number of frames with other fine-tuning strategies.

| Method | $d_{bottle}$ | Pre-training | # Frames | SSv2 | | HMDB51 | |
|---|---|---|---|---|---|---|---|
| | | | | # Params | Top-1 (%) | # Params | Top-1 (%) |
| Full-tuning | - | Kinetics 400 | 8 | 87.82M | **50.99** | 87.69M | 68.07 |
| Tune FC Layer | - | Kinetics 400 | 8 | 0.18M | 24.13 | 0.05M | **71.28** |
| BitFit (Zaken et al., 2022) | - | Kinetics 400 | 8 | 1.29M | 45.94 | 1.11M | 68.26 |
| AdaptFormer-Swin (Chen et al., 2022) | 64 | Kinetics 400 | 8 | 1.73M | 40.80 | 1.61M | 68.66 |
| Prefix-tuning (Li & Liang, 2021) | 128 | Kinetics 400 | 8 | 6.57M | 39.46 | 6.40M | 56.13 |
| Our Swin-BAPAT (w/o Adapter) | 32 | Kinetics 400 | 8 | 1.35M | 46.26 | 1.17M | 69.51 |
| Our Swin-BAPAT (w/o Adapter) | 64 | Kinetics 400 | 8 | 2.51M | 49.23 | 2.34M | **71.34** |
| Our Swin-BAPAT (w/o Adapter) | 128 | Kinetics 400 | 8 | 4.83M | 52.57 | 4.65M | 70.56 |
| Our Swin-BAPAT (w/o Adapter) | 256 | Kinetics 400 | 8 | 9.45M | **52.71** | 9.27M | 70.23 |
| Our Swin-BAPAT | 32 | Kinetics 400 | 8 | 2.91M | 49.63 | 2.74M | 68.20 |
| Our Swin-BAPAT | 64 | Kinetics 400 | 8 | 4.07M | 51.80 | 3.89M | 70.10 |
| Our Swin-BAPAT | 128 | Kinetics 400 | 8 | 6.38M | 53.36 | 6.20M | **71.93** |
| Our Swin-BAPAT | 256 | Kinetics 400 | 8 | 11.00M | **53.98** | 10.83M | 69.64 |
| Our Swin-BAPAT | 256 | Kinetics 400 | 8 | 11.00M | 53.98 | 10.83M | 69.64 |
| Our Swin-BAPAT | 256 | Kinetics 600 | 8 | 11.00M | **54.06** | 10.83M | **69.90** |
| Our Swin-BAPAT | 256 | ImageNet-22K | 8 | 11.00M | 43.56 | 10.83M | 59.89 |
| Our Swin-BAPAT | 128 | Kinetics 400 | 8 | 6.38M | 53.36 | 6.20M | 71.93 |
| Our Swin-BAPAT | 128 | Kinetics 400 | 16 | 6.38M | **63.14** | 6.20M | **75.67** |

etc. Our PATT module is convenient to be applied to other Transformer-based models. Hence, we respectively adopt ViT-B models from MAE (He et al., 2022b) and VideoMAE (Tong et al., 2022) to conduct further comparison on video and image datasets, which follows the self-supervised pre-training setting[2] in Chen et al. (2022) except that the batch size is set to $256$ instead of $1,024$.

**Baselines:** We mainly compare our method Swin-BAPAT with three baselines as follows: (1) Full-tuning: set all the parameters learnable and tune the whole model initiated with the pre-trained weights. (2) Tune FC layer: tune the last fully connected layer and freeze pre-trained parameters of the whole backbone model. (3) AdaptFormer-Swin: method introduced by Chen et al. (2022) that adds a parallel adapter to the MLP module in each block of the backbone model. (4) Prefix-tuning: the direct implementation of prefix-tuning used in NLP as defined in Eq. 5. (5) BitFit: by tuning the bias of the backbone model together with the FC layer.

## 3.2 THE EFFECT OF DIFFERENT PETL TECHNIQUES

Table 2 shows the results of different PETL techniques. From the results of four baseline methods, full-tuning performs the best for the large-scale dataset SSv2, whereas tuning the FC layer achieves superior performance over other PETL techniques on HMDB51. This is due to the fact that downstream tasks with relatively larger scale datasets are more parameter hungry for good convergence. On the contrary, small datasets can make good use of the knowledge from the source domain with slight effort of adaption via an FC layer. Here, a question regarding the effect of this FC layer when using it together with other PETL techniques has not been investigated. As this FC layer having small amount of tunable parameters can already make a big difference, performing better than full-tuning and other PETL techniques and rendering them not effective for small-scale datasets. As such, we further examine this question in Section A.1.

We test different amount of parameters adjusted by $s_{bottle}$, taking its values to 32, 64, 128 and 256. The second and third groups (without or with Adapter, respectively) of results in Table 2 shows that larger values of $s_{bottle}$ can benefit the fine-tuning with slightly more overhead of parameters on large-scale datasets such as SSv2. All results of our Swin-BAPAT outperform the state-of-the-art AdaptFormer-Swin with a big margin (using the smallest value $s_{bottle} = 32$ can improve AdaptFormer-Swin by almost 25%). While without using Adapter, our method still outperforms baselines AdaptFormer-Swin and BitFit with roughly similar amount of parameters. When $s_{bottle}$ is larger than 64, our Swin-BAPAT starts to perform better than full-tuning on both datasets with proper parameter-accuracy trade-off, validating the effectiveness of our Swin-BAPAT for PETL.

---

[2]https://github.com/ShoufaChen/AdaptFormer/blob/main/PRETRAIN.md

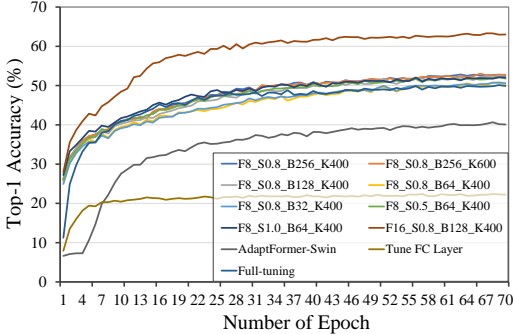

Figure 4: Top-1 accuracy of different settings on SSv2 throughout training process. F: frame, S: scalar, B: $d_{bottle}$, K: pre-training domain.

Table 3: Top-1 accuracy (%) using different scalar values on two datasets: SSv2 and HMDB51. The $d_{bottle}$ is set to 128; pre-training is based on Kinetics 400.

| Scalar $s$ | SSv2 | HMDB51 |
|---|---|---|
| Full-tuning | 50.99 | 71.28 |
| Tune FC Layer | 24.13 | 68.07 |
| AdaptFormer-Swin | 40.80 | 68.66 |
| $s = 0.2$ | 47.46 | 69.38 |
| $s = 0.5$ | 52.84 | 71.87 |
| $s = 0.8$ | **53.36** | **71.93** |
| $s = 1.0$ | 53.29 | 70.89 |

### 3.3 THE EFFECT OF DIFFERENT PRE-TRAINING DOMAINS

The knowledge from the pre-trained model is learned from the source domain. We test two different models pre-trained on large-scale datasets: Kinetics 400, Kinetics 600, and ImageNet-22K. Findings show that both two models pre-trained on such large-scale datasets can benefit our proposed PETL strategy with the latter being slightly more significant (see the third group of comparison in Table 2). This is due to the fact that Kinetics 600 is larger than its 400 version and brings more knowledge to the pre-trained model, benefiting more downstream tasks. However, image-based pre-training cannot perform as good as video-based pre-training due to the larger domain gap.

### 3.4 THE EFFECT OF DIFFERENT VIDEO INPUT SIZE

We also test whether our method is robust to increased number of input video frames. It is worth noting that larger number of input video frames usually can bring more spatial temporal information, benefiting data-driven models to learn more distinguishable features while keeping the model size remaining the same. The last group of comparisons in Table 2 shows that using double-sized video input (i.e., 16 frames) can greatly improve the performance of action recognition on both small and large-scale datasets. The improvements (increased 9.78% from 53.36% to 63.14% on SSv2, and 3.74% from 71.93% to 75.67% on HMDB51) are more significant than other factors such as $d_{bottle}$ and pre-training domain (around 1% to 2%). The top line in Figure 4 visualizes the significant effect of increasing the number of input video frames. These results suggest that our Swin-BAPAT can be promising for increased frames of video input.

### 3.5 THE EFFECT OF DIFFERENT SCALE OF PATT

Recall that the effect of our PATT on pre-trained models can be adjusted by the variable $s$ in Eq. 10. Table 3 shows that adopting the value of 0.8 can deliver consistent best performances on both datasets SSv2 and HMDB51 under our experimental setting. Smaller values of $s$ will quantitatively reduce the effect of our PATT module on the knowledge transfer while large values will increase the effect of our PATT module. The good performance achieved via taking an effective scale of 0.8 indicates that our PATT module plays an important role in the knowledge transfer. However, even larger values over 0.8 can affect the importance of original knowledge thereof the pre-trained model. Hence, proper valued scalar $s$ is essential for balancing the role of PATT and

Table 4: Ablation of different implementation positions of PATT defined in Eq. 10, e.g., Ours (**K**, **V**) indicates inserting PATT to the query and key of 3DSW-MSA modules. Pre-training on Kinetics 600. $d_{bottle}$ is set to 128; Scalar $s$ is set to 0.8.

| Method | SSv2 | | HMDB51 | |
|---|---|---|---|---|
| | # Params | Top-1 | # Params | Top-1 |
| Full-tuning | 87.82M | 50.99 | 87.69M | 68.07 |
| Concat (**K**, **V**) | 6.38M | 15.61 | 6.20M | 20.98 |
| No $Z^{l-1}$ (**K**, **V**) | 8.74M | 51.06 | 8.56M | 67.41 |
| Ours (**Q**, **K**) | 6.38M | 45.49 | 6.20M | 68.92 |
| Ours (**K**, **V**) | 6.38M | **53.38** | 6.20M | 71.41 |
| Ours (**Q**, **V**) | 6.38M | 53.24 | 6.20M | **71.74** |
| Ours (**Q**, **K**, **V**) | 7.93M | 53.23 | 7.63M | 69.57 |

pre-trained backbone model. Note this can be a learnable parameter upon specific implementation, here we empirically verified the effect of the scalar.

### 3.6 THE EFFECT OF DIFFERENT METHODS YIELD FROM V-PETL

We have argued that, especially for relative large downstream datasets, the position and the amount of trainable parameters are important for parameter-efficient transfer learning in Section 2.4. The proposed Swin-BAPAT is one of instantiated models from the V-PETL framework regarding the insert position of our PATT. Other instantiations can be inserted into different positions such as query, key, and value of the attention module. We further instantiate other variations of our Swin-BAPAT by inserting PATT to different positions. Table 4 shows the results. Findings show that inserting to the value position of 3DSW-MSA can contribute more than inserting to other two positions. While inserting to query of key makes little difference for the performance. This is due to the fact that query and key make the calculation of the attention mask. Hence, inserting either one of them will lead to a similar effect. On one hand, these results, to some extent, justify the original design of prefix-tuning that bring learnable prefix to key and value of the attention module. On the other hand, it indicates that our claim regarding the unified view of PETL for visual tasks is reasonable. In Table 4, we also ablate the designs of PATT regarding concatenating $K_p$ and $V_p$ (i.e., Concat [$\mathbf{K}$, $\mathbf{V}$]), and using trainable parameters to generate $K_p$ and $V_p$ (i.e., No $Z^{l-1}$ [$\mathbf{K}$, $\mathbf{V}$]).

### 3.7 COMPARISON ON VARIED TASKS VIA SELF-SUPERVISED PRE-TRAINED MODELS

Table 5 shows the comparison with AdaptFormer-64 (Chen et al., 2022) and VPT (Jia et al., 2022) on both image- and video-based downstream tasks. Our method ViT-BAPAT still shows promising parameter-accuracy trade-off via much smaller batch size, which is more convenient for reproduction on the general single server with 8 GPUs. The underperformance on SSv2 (better than full-tuning) can be due to the smaller batch size as SSv2 is much larger than other compared datasets and can be more relying on larger batch size. In real-world application scenarios, small dataset can be the more common case, which confirms our contributions.

Table 5: Comparison of Top-1 accuracy via ViT-B models from MAE and VideoMAE pre-trained with self-supervised learning for image and video datasets, respectively.

| Method | Avg. Params (M) | Image | | | Video | |
| --- | --- | --- | --- | --- | --- | --- |
| | | CIFAR-100 | SVHN | Food-101 | SSv2 | HMDB51 |
| Full-tuning | 86.04 (100%) | 85.90 | 97.67 | 90.09 | 53.97 | 46.41 |
| Tune FC Layer | 0.07 (0.08%) | 69.83 (-16.07) | 66.91 (-30.76) | 69.74 (-20.35) | 29.23 (-24.74) | 49.84 (+3.43) |
| VPT (Jia et al., 2022) | 0.08 (0.09%) | 82.44 (-3.46) | 94.02 (-3.65) | 82.98 (-7.11) | 43.73 (-10.24) | 52.67 (+6.26) |
| AdaptFormer-64 | 1.26 (1.46%) | 85.90 (0.00) | 96.89 (-0.78) | 87.61 (-2.48) | 59.02 (+5.05) | 55.69 (+9.28) |
| Our ViT-BAPAT-32 | 2.13 (2.47%) | 86.29 (+0.39) | 97.18 (-0.49) | 87.37 (-2.72) | 57.78 (+3.81) | 57.18 (+10.77) |
| Our ViT-BAPAT-64 | 3.02 (3.51%) | 86.35 (+0.45) | 97.18 (-0.49) | 87.53 (-2.56) | 57.55 (+3.58) | 57.18 (+10.77) |
| Our ViT-BAPAT-128 | 4.79 (5.56%) | 86.47 (+0.57) | 97.28 (-0.39) | 87.66 (-2.43) | 56.97 (+3.00) | 57.70 (+11.29) |
| Our ViT-BAPAT-256 | 8.33 (9.68%) | 86.55 (+0.65) | 97.24 (-0.43) | 87.68 (-2.41) | 56.53 (+2.56) | 57.31 (+10.90) |

## 4 CONCLUSION

In this paper, we introduced a V-PETL framework for exploiting good parameter-accuracy trade-off around adapting video-based pre-trained large models to downstream tasks. Our Swin-BAPAT method derived from the V-PETL with a variation of prefix-tuning known as PATT can effectively bring good parameter-accuracy trade-off on downstream tasks. The proposed PATT can be easily plugged to the attention module of other transformer-like models. Meanwhile, the amount of trainable parameter can be easily adjusted by the parameter $d_{bottle}$. With small amount overhead on trainable parameters, our method performs significantly better than state-of-the-art method AdapFormer-Swin and full-tuning on the datasets SSv2 and HMDB51 via small batch size, validating our contribution to the literature of PETL. In the future we will test our proposed model on more action recognition datasets surveyed in Sun et al. (2022) under more learning regimes such as zero/few-shot learning, active learning and continual learning with other pre-training methods such as visual-language models. We will also explore other backbone models, activation functions for PATT, and PETL techniques such as LoRA for visual tasks.

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

# A  APPENDIX

## A.1  THE EFFECT OF FC LAYER FOR SMALL SCALE DOWNSTREAM TASKS

Table 6: Results of with or without tuning the FC layer on the small scale dataset HMDB51.

| Method | $d_{bottle}$ | Pre-training | # Frames | with FC layer | | without FC layer | |
| --- | --- | --- | --- | --- | --- | --- | --- |
| | | | | # Params | Top-1 (%) | # Params | Top-1 (%) |
| Our Swin-BAPAT | 32 | Kinetics 400 | 8 | 2.79M | 65.97 | 2.74M | 68.20 |
| Our Swin-BAPAT | 64 | Kinetics 400 | 8 | 3.94M | 67.28 | 3.89M | 70.10 |
| Our Swin-BAPAT | 128 | Kinetics 400 | 8 | 6.25M | 66.75 | 6.20M | **71.93** |
| Our Swin-BAPAT | 256 | Kinetics 400 | 8 | 10.88M | **67.67** | 10.83M | 69.64 |
| Our Swin-BAPAT | 256 | Kinetics 400 | 8 | 10.88M | **67.67** | 10.83M | 69.64 |
| Our Swin-BAPAT | 256 | Kinetics 600 | 8 | 10.88M | 67.41 | 10.83M | **69.90** |
| Our Swin-BAPAT | 128 | Kinetics 400 | 8 | 6.25M | 66.75 | 6.20M | 71.93 |
| Our Swin-BAPAT | 128 | Kinetics 400 | 16 | 6.25M | 70.56 | 6.20M | 75.67 |
| Our Swin-BAPAT | 128 | Kinetics 400 | 32 | 6.25M | **74.82** | 6.20M | **76.46** |

For the small dataset HMDB51, due to the good parameter-accuracy trade-off achieved by fine-tuning the FC layer only, adding the FC layer cannot bring extra improvement to our proposed method. Without sufficient taining data, full-tuning also cannot perform well (see results in Table 2). As such, small datasets do not need to rely on large models but can make use of large models with light transfer. Instead, without tuning the FC layer, our Swin-BAPAT can perform better than fine-tuning the FC layer with small amount of extra trainable parameters (see results in Table 6), validating the good parameter-accuracy trade-off of our method.

