# OpenReview forum: "Towards a Unified View on Visual Parameter-Efficient Transfer Learning"
_ICLR.cc/2023/Conference — Submitted to ICLR 2023_

### Official Review · Reviewer_EH6p · 2022-10-22

**Confidence:** 5
**Correctness:** 3
**Technical Novelty And Significance:** 2
**Empirical Novelty And Significance:** 2
**Recommendation:** 5

**Clarity, Quality, Novelty And Reproducibility:**

- Clarity, Quality:
I think the paper presentation is clear, and the proposed method is easy to understand.

- Novelty:
Problem novelty: ST-Adapter [1] and AdapterFormer [2] have addressed the same problem settings.
Method novelty: Since the proposed method is based on Prefix tuning [3], the way they introduce the learnable parameters have been explored by the prior work. The only interesting point is they further reduce the parameter usage by using a autoencoder-like module rather than additional linear layers to produce keys/values, although this method is similar to LoRA [4].

Ref:
[1] "ST-Adapter: Parameter-Efficient Image-to-Video Transfer Learning", Pan et al., NeurIPS 2022
[2] "AdaptFormer: Adapting Vision Transformers for Scalable Visual Recognition", Chen et al., NeurIPS 2022.
[3] "Prefix-Tuning: Optimizing Continuous Prompts for Generation", Li et al.
[4] "LoRA: Low-Rank Adaptation of Large Language Models", Hu et al.

**Strength And Weaknesses:**

**Strengthes**
- The paper presentation is good, and the authors clearly show how the proposed method is designed with easy-to-understand figures (Figure 2 and 3).
- The authors presented several analyses on the hyper-parameters (Table 2, 3, 4, and 5).

**Weaknesses**
- Paper title does not properly describe the paper content: Since the a large portion of the paper deals with the video data and video recognition tasks, I would suggest the author either include more different types of vision tasks or narrow the title to "for video recognition tasks".


- In section 2.3, "prefix implementation in NLP can be regarded as providing conditional information for the downstream tasks, which is similar with the pre-training process where words are masked". The sentence is confusing to me, since the connection between the prefix tuning and masking text pretraining does not look similar to me. I would suggest to elaborate this motivation and provide a more reasonable motivation why the prefix tuning is suitable for video tasks.

- CLIP is not one of parameter-efficient methods, so it should be removed from Section 2.2

- Less convincing experiments:

1. Missing an important baseline - ST-Adapter [1]: Since the paper addresses the video tasks, one important baseline (ST-Adapter [1] ) should be included in the experiment. ST-Adapter present several video benchmarks, including SSv2 and K-400 datasets with different pretrained weights.

2. *Comparison to AdapterFormer*: Although the authors compare AdapterFormer, there are still some concerns in the comparison. First, the experimental setup of AdapterFormer paper is different from the proposed method, since AdapterFormer uses different settings (pretrained on IN-21k and adapt to SSv2). I would suggest the authors also include their experimental setups (same backbone, same pretraining datasets, and same downstream tasks) to provide a complete comparison. It is unclear whether the baseline is properly tuned and well reproduced. Also, the authors show the number of parameters of AdapterFormer is only 1.73M, while the AdapterFormer, which is based on adapter, can certainly be improved by increasing the hidden state size of adapters.

3. Experiment setups are still limited. The authors only show a single model architecture, video SwinTransformer, and I would suggest the authors extend to other model architecture. I would be interested in how the proposed method performs under ViT with ImageNet-21k perform on K400.

Ref:
[1] "ST-Adapter: Parameter-Efficient Image-to-Video Transfer Learning", Pan et al., NeurIPS 2022

**Summary Of The Paper:**

The goal of this paper is to reduce the parameter usage when adapting pretrained models to video tasks. To this end, the authors propose a new parameter-efficient method, which adds the learnable parameters to generate additional keys and values. Unlike the prior Prefix-tuning, the authors further reduce trainable parameters by introducing an autoencoder-like modules for generating keys and values. In the experiment, the authors show the improvements when adapting a Kinetics 400 pretrained models to SSv2 and HMDB51 datasets.

**Summary Of The Review:**

While the authors propose a new parameter-efficient method for video classification tasks, there exist some concerns and questions in the experiment sections. More experimental setups and baselines should be considered for a more complete and convincing empirical results. Given the current status of the paper, I would suggest the score "marginally below the acceptance threshold".

---

> ### Author Response · Authors · 2022-11-19
> **Response to Reviewer EH6p (Part 1/3)**
>
> Dear Reviewer EH6p,
>
> Thank you for your valuable comments. We conducted more experiments and included all discussions and results in our revised manuscript (blue content in the main text).
>
> **Q1 Paper title.**
> Out initial concern is on the small batch size setting for video-based tasks while image data has less of this small batch-size concern. For the complete comparison raised in **Q4**, we conduct experiments by using the same experimental setups (same backbone, same pretraining datasets, and same downstream tasks). Specifically, we use the architectures: ViT-B models from MAE [a] and VideoMAE [b] for three image and two video datasets, respectively. Self-supervised pre-training is used, which follows the main paper settings of AdaptFormer. The results are as follows (also see Table 5 of the revised version), which can draw consistent conclusion in the initial version (i.e., promising parameter-accuracy trade-off):
>
> | Method | Avg. Params (M) | CIFAR-100  |  SVHN  |  Food-101  |  SSv2  |  HMDB51  |
> | :--- | :----: | :----:  | :----:  | :----: | :----:  | :----:  |
> | Full-tuning | 86.04 (100%) | 85.90 | 97.67 | 90.09 | 53.97 | 46.41 |
> | Tune FC layer | 0.07 (0.08%) | 69.83 (-16.07) | 66.91 (-30.76) | 69.74 (-20.35) | 29.23 (-24.74) | 49.84 (+3.43) |
> | VPT | 0.08 (0.09%) | 82.44 (-3.46) | 94.02 (-3.65) | 82.98 (-7.11) | 43.73 (-10.24) | 52.67 (+6.26) |
> | AdaptFormer-64 | 1.26 (1.46%) | 85.90 (0.00) | 96.89 (-0.78) | 87.61 (-2.48) | 59.02 (+5.05) | 55.69 (+9.28) |
> | Our ViT-BAPAT-32 | 2.13 (2.47%) | 86.29 (+0.39) | 97.18 (-0.49) | 87.37 (-2.72) | 57.78 (+3.81) | 57.18 (+10.77) |
> | Our ViT-BAPAT-64 | 3.02 (3.51%) | 86.35 (+0.45) | 97.18 (-0.49) | 87.53 (-2.56) | 57.55 (+3.58) | 57.18 (+10.77) |
> | Our ViT-BAPAT-128 | 4.79 (5.56%) | 86.47 (+0.57) | 97.28 (-0.39) | 87.66 (-2.43) | 56.97 (+3.00) | 57.70 (+11.29) |
> | Our ViT-BAPAT-256 | 8.33 (9.68%) | 86.55 (+0.65) | 97.24 (-0.43) | 87.68 (-2.41) | 56.53 (+2.56) | 57.31 (+10.90) |
>
> Given that image datasets are tested, we will keep the current title.
>
> **Discussion**
>
> Based on the direct comparison with AdaptFormer on varied visual tasks, our method still shows promising parameter- accuracy trade-off via much smaller batch size, which is more convenient for reproduction on the general single server with 8 GPUs. We will test our method when a 64-GPU cluster is available for an absolute fair comparison. The underperformance on SSv2 (better than full-tuning) can be due to the smaller batch size as SSv2 is much larger than other compared datasets and can be more relying on larger batch size. In real-world application scenarios, small dataset can be the more common case, which confirms our contribution to the community.
>
> **Q2 Description in Section 2.3, provide a more reasonable motivation why the prefix tuning is suitable for video tasks.**
>
> Thank you for the suggestion. We revised the first paragraph of Section 2,3 as follows:
>
> “The prefix implementation in NLP Li & Liang (2021); He et al. (2022a) c can be regarded as prepending contextual information for downstream tasks, which is similar with the pre-training process aiming to predict masked words in the process of an inner loop [c]. Considering the pre-training process of pure vision models, such direct implementation might not make sense for visual tasks. Although such autoregressive pre-training has been conducted in visual domain [a,b] but adding prefix for a sentence input in NLP can be structurally different with the visual domain. Specifically, masked pixels in image or video data cannot be regarded as some word level semantic information (e.g., a subject or an action) as in the NLP.”
>
> **Q3 In Section 2.2, CLIP is not one of parameter-efficient methods.**
>
> Thank you for the correction. We revised the last paragraph of Section 2.2 as follows:
>
> “Others: Other PETL techniques include ST-Adapter Pan et al. (2022), LoRA (Hu et al., 2022), and
> BitFit (Zaken et al., 2022). ST-Adapter mainly adapts image-text models pre-trained on large scale
> datasets such as 400M image-text pair proposed by CLIP (Radford et al., 2021) and the IG-3.6B
> used by SWAG (Singh et al., 2022) to video understanding downstream tasks, which matches and
> even outperforms full-tuning. LoRA approximates the optimization process by injecting learnable
> low-rank matrices into the attention module. This method does not show superior performance for
> NLP tasks in terms of parameter efficiency. Hence, we do not prioritize this direction in this work.
> BitFit only tunes the bias terms of the backbone models, making it very parameter-efficient.”

---

> > ### Author Response · Authors · 2022-11-19
> > **Response to Reviewer EH6p (Part 2/3)**
> >
> > **Q4 Experiments**
> >
> > **Q4.1 Missing an important baseline - ST-Adapter**
> >
> > Thank you for pointing the ST-Adapter paper. ST-Adapter is published in NIPS-22 on 1 Nov 2022 (after over 1 month of our ICLR submission on 21 Sept 2022). Hence, as you probably know, ST-Adapter should be considered as a contemporaneous work. We just noticed that ST-Adapter’s first arXiv version was available on 27 Jun 2022 (within three months of our ICLR-23 submission). **According to the [ICLR-23 Reviewer Guide](https://iclr.cc/Conferences/2023/ReviewerGuide), authors may be excused for not knowing about papers published within the last four months (on or after 28 May 2022) or not published in peer-reviewed conference proceedings or journals, which includes papers exclusively available on arXiv**. Another contemporaneous NIPS-22 work, AdaptFormer’s first arXiv version was released on 26 May 2022 (one month earlier than ST-Adapter, but still not yet published). Hence, our current version mainly follows the experimental setting (using SGD and without augmentation) of AdaptFormer concerning the situation of insufficient computing resources. The different experimental settings of AdaptFormer and ST-Adapter are as follows:
> >
> > | Method  | AdaptFormer  | ST-Adapter |
> > | :---  | :----  |  :----  |
> > | Frames |  8 | 8 $\times$ 3|
> > | Optimizer | SGD | AdamW |
> > | Training resize | No | RandomResizedCrop |
> > | Sampling | Uniform | Dynamics |
> > | Augmentation | MultiScaleCrop | RandAugment |
> >
> > By the date of ICLR’s first round revision, ST-Adapter has not released the [code on GitHub]( https://github.com/linziyi96/st-adapter). We would be happy to see their code and conduct a further comparison with ST-Adapter in the future.
> >
> > **Q4.2 Comparison to AdapterFormer.**
> >
> > **Different settings with AdaptFormer**:
> > The only different setting with AdaptFormer is the smaller batch size. AdaptFormer has source code released. We will also release our source code to verify the consistency in implementation. The main experiments in AdaptFormer is using self-supervised pre-training. We have conducted experiments by following this setting and the results are in the response to Q1.
> >
> > According to the result of AdaptFormer pre-training with IN-21k actually do not perform better than pre-training with video datasets such as Kinetics 600 (46.06% vs 60.18%). We also used Swin-B pre-trained with ImageNet-22K for experiments on SSv2 and HMDB51. The results also show that video pre-trained weights can perform better than image pre-trained weights, as follow:
> >
> > | Method | $d_{bottle}$ | Pre-training | # Params | SSv2 (Top-1) | # Params | HMDB51 (Top-1) |
> > | :--- | :----: | :----:  | :----:  | :----: | :----:  | :----:  |
> > | Our Swin-BAPAT | 256 | Kinetics 400 | 11.00M | 53.98% | 10.83M | 69.64% |
> > | Our Swin-BAPAT | 256 | Kinetics 600 | 11.00M | 54.06% | 10.83M | 69.90% |
> > | Our Swin-BAPAT | 256 | ImageNet-22K | 11.00M | 43.56% | 10.83M | 59.89% |
> >
> > **Impact of hidden state size**:
> > Although on some datasets, increase the hidden state size of adapters can improve AdaptFormer’s performance. However, on SSv2, AdaptFormer cannot be improved on by increasing the hidden state size of adapters. AdaptFormer did ablation on the hidden state size of adapters (see Table 3[a] of the AdaptFormer paper), where the best practice for $\hat{d}$ is 64. It has trivial difference when $\hat{d}$ is larger than 64 and even declined when it is set to 256. We paste the AdaptFormer’s ablation results of increased hidden state size as follows:
> >
> > | Middle Dim | Params | SSv2 Top1 |
> > | :---  | :----:  | :----:  |
> > | 1 | 0.16M | 50.03% |
> > | 4 | 0.22M | 54.7% |
> > | 16 | 0.44M | 57.62% |
> > | 32 | 0.73M | 58.27% |
> > | 64 | 1.32M | 59.02% |
> > | 128 | 2.51M | 58.95% |
> > | 256 | 4.87M | 58.87% |
> > | 512 | 9.59M | 58.98% |
> >
> > **Discussion**:
> > According to the results in AdaptFormer and ST-Adapter, full-tuning the ViT with ImageNet-21k can perform 40% via 128 batch size on SSv2 while 41.5% via 1,024 batch size. [d] also shows that larger batch size (512 or 1,024 can benefit the performance). Our contribution is based on very reproducible setting (i.e., using smaller batch size) for the video recognition task under the background that existing works such as AdaptFormer and ViTMAE rely on huge computational resources with a batch size of 1,024 for training. More importantly, when the batch size is small, their performance will decline with a considerable margin. AdaptFormer also reported the results using Swin-B in its supplementary. Tuning 1.25M parameters, AdaptFormer can achieve 54.09% by adapting the Kinetics 600 to SSv2 via the Swin-B. While, with our proposed PATT module, AdaptFormer-Swin can be improved from 40.80 to 54.06 with slightly more parameters (less than 10M) and the far smaller batch size of 64.

---

> > > ### Author Response · Authors · 2022-11-19
> > > **Response to Reviewer EH6p (Part 3/3)**
> > >
> > > **Q4.3 Results on K400 using ViT pre-trained with ImageNet-21k.**
> > >
> > > Thank you for this good suggestion. Following the experimental setting of AdaptFormer, we extend our method using ViT pre-trained with ImageNet-21k and test it on Kinetics 400. The results at the 40th epoch is 63.6% under our experimental setting. We will report the final result after the training is complete. Again, due to the different experimental setting with ST-Adapter  (see response to **Q4.1**), it cannot be directly compared with the results in ST-Adapter. We will address this with more experiments in the future.
> > >
> > > **Q5 Recap the Novelty and Contribution**
> > >
> > > Thank you for the remark regarding the novelty. After revision based on your valuable comments, our impact can be from three perspectives: technically sound, convenient for reproduction, and rigorous experiments with promising results as follows:
> > >
> > > * Our work is the first to implement prefix-tuning for the visual domain. AdaptFormer (Chen et al., 2022) is relatively easier to implement since it is just plug-in the Adapter to the MLP without considering the different data structures of NLP and visual domains. As far as we know, AdaptFormer can be regarded as a direct and easy implementation case of (He et al., 2022) given that the first arXiv version of (He et al., 2022) is on 8 Oct 2021 while NIPS-22’s submission date is over seven months later (16 May 2022). Due to the barrier caused by the different data structures, no previous work has investigated the effect of prefix-tuning for visual tasks. Based on the unsatisfied results of our initial attempt at prefix-tuning that tasks the inspiration of (He et al., 2022), we modify the form of prefix-tuning via a different calculation of $K_p$ and $V_p$ as described in the whole content of Section 2.3.
> > >
> > > * Taking the reality that a larger batch size (i.e., 512 or 1024) usually leads to better performance than implementing via a smaller batch size of 64 [a], our method can outperform compared baselines such as BitFit, AdaptFormer-Swin, and Prefix-tuning with a similar number of parameters under the small batch size setting as follows:
> > >
> > > | Method | $d_{bottle}$  | # Params | SSv2 (Top-1) | # Params | HMDB51 (Top-1) |
> > > | :--- | :----: | :----:  | :----:  | :----: | :----:  |
> > > | Full-tuning | - | 87.82M | 50.99% | 87.69M | 68.07% |
> > > | Tune FC Layer | - | 0.18M | 24.13% | 0.05M | 71.28% |
> > > | **BitFit** | - | 1.29M | 45.94% | 1.11M | 68.26% |
> > > | AdaptFormer-Swin (Adapter) | 64 | 1.73M | 40.80% | 1.61M | 68.66% |
> > > | Our Swin-BAPAT (w/o Adapter) | 32 | 1.35M | 46.26% | 1.17M | 69.51% |
> > > | Our Swin-BAPAT | 32 | 2.91M | 49.63% | 2.74M | 68.20% |
> > >
> > > * Our revision incorporates more rigorous and extensive experiments on different visual tasks: image and video. Following the experimental setting of AdaptFormer except that we use a smaller batch size, our method still shows a good parameter-accuracy trade-off (see results in responses to Q1 with discussion).
> > >
> > > ------
> > >
> > > **References**
> > >
> > > [a] He et al. Masked autoencoders are scalable vision learners, CVPR 2022
> > >
> > > [b] Tong et al. Videomae: Masked autoencoders are data-efficient learners for self-supervised video pre-training, NIPS 2022
> > >
> > > [c] Brown et al. Language Models are Few-Shot Learners, NIPS 2020
> > >
> > > [d] Chen et al. Exploring Simple Siamese Representation Learning, CVPR 2021

---

> > > > ### Author Response · Authors · 2022-12-09
> > > > **Results on Kinetics 400 and Look Forward to Your Feedback**
> > > >
> > > > Dear Reviewer EH6p,
> > > >
> > > > The results of using ViT pre-trained with ImageNet-21k and testing on Kinetics 400 are as follows:
> > > >
> > > > | Method | # Params | Kinetics 400 (Top-1) |
> > > > | :--- | :----: | :----:  |
> > > > | Full-tuning | 86.53M | 60.59% |
> > > > | AdaptFormer |1.52M | 70.91% |
> > > > | Our ViT-BAPAT-128 | 4.77M | 71.63% |
> > > >
> > > > As well as the results of other added experiments on image and video tasks, the above results also indicate that our method can improve the performance of AdaptFormer with a good parameter-accuracy trade-off.
> > > >
> > > > We sincerely look forward to your reply and further feedback. Thank you.
> > > >
> > > > Best regards,
> > > >
> > > > Authors

---

### Official Review · Reviewer_uEYB · 2022-10-25

**Confidence:** 4
**Correctness:** 4
**Technical Novelty And Significance:** 2
**Empirical Novelty And Significance:** 3
**Recommendation:** 5

**Clarity, Quality, Novelty And Reproducibility:**

The paper is generally well-written. The novelty of the work on the other hand is limited due to the overlap to the prior work (He et al., 2022).

**Strength And Weaknesses:**

Strength:
- The motivation for applying parameter-efficient learning to videos is sound and reasonable.
- Ablation studies and empirical results are strong and convincing.

Weakness:
1. Novelty
The ideas presented in the paper–steering the Transformer via a prefix-tuning-like modification to MHSA and adaptor-like modification to the MLP–are already explored by He et al., 2022, which leads to a very similar model. Although there are some differences in details, such as how we incorporate prefix-tuning in self-attention, these are rather minor and can be considered as the special case of the prior work (He et al., 2022). Although I agree that extending the observations to videos can be valuable, the impact is still not significant considering that success in the NLP domain tends to easily extend to the visual domains.


2. Experiments
- Although the same method is applicable to images, the paper is evaluated only on videos. To understand the benefits and limitations of the proposed method better, it would be great to see the results in image downstream tasks as in the prior works (e.g., Chen et al., 2022).

- The comparisons to AdaptFormer (Chen et al., 2022) in Table 2 need more careful considerations. It is not easy to directly compare the two methods (AdaptFormer and the proposed method) since the number of parameters are different (AdaptFormer uses fewer parameters). Considering the results from Chen et al., 2022 that the performance of AdaptFormer also increases proportionally to the number of parameters, it would be more desirable to roughly match the parameters and compare the performance in multiple scales.


3. Presentations
- There are some minor presentation issues.
  1) the first dimension of B (in the line above Eq.(2)) should be t->p.
  2) in Table 1, the first two rows should be switched.

- It would also be interesting to consider comparisons with much more parameter-efficient alternatives such as BitFit (Ben-Zaken et al., 2022).


**Summary Of The Paper:**

Motivated by recent studies on parameter-efficient learning of Transformers in NLP (He et al., 2022) and vision domain (Jia et all., 2022), the authors proposed to extend them to video models. Numerous ablation studies are conducted to find the best configuration for incorporating parameter efficient approaches in video Transformers, which turned out to be consistent with the previous observation (He et al., 2022) where the self-attention and MLP layers in Transformers are modified with a prefix-tuning-like approach and adaptor, respectively. The proposed method is evaluated in two video downstream tasks and demonstrated encouraging results.

**Summary Of The Review:**

Overall, I lean towards rejecting this paper since the ideas and results from the paper largely overlap with the prior work (He et al., 2022) and the experiments only demonstrate the results in videos.

---

> ### Author Response · Authors · 2022-11-19
> **Response to Reviewer uEYB (Part 1/2)**
>
> Dear Reviewer uEYB,
>
> Thank you for your comments. We recap the novelty of our work and conducted more extensive experiments below. We have included all discussions and results in our revised manuscript (blue content in the main text).
>
> **Q1 Recap the significant impact of our work.**
>
> Thank you for the remark regarding the novelty. After revision based on your valuable comments, our impact can be from three perspectives: technically sound, convenient for reproduction, and rigorous experiments with promising results as follows:
>
> * Our work is the first to implement prefix-tuning for the visual domain. AdaptFormer (Chen et al., 2022) is relatively easier to implement since it is just plug-in the Adapter to the MLP without considering the different data structures of NLP and visual domains. As far as we know, AdaptFormer can be regarded as a direct and easy implementation case of (He et al., 2022) given that the first arXiv version of (He et al., 2022) is on 8 Oct 2021 while NIPS-22’s submission date is over seven months later (16 May 2022). Due to the barrier caused by the different data structures, no previous work has investigated the effect of prefix-tuning for visual tasks. Based on the unsatisfied results of our initial attempt at prefix-tuning that tasks the inspiration of (He et al., 2022), we modify the form of prefix-tuning via a different calculation of $K_p$ and $V_p$ as described in the whole content of Section 2.3.
>
> * Taking the reality that a larger batch size (i.e., 512 or 1024) usually leads to better performance than implementing via a smaller batch size of 64 [a], our method can outperform compared baselines such as BitFit, AdaptFormer-Swin, and Prefix-tuning with a similar number of parameters under the small batch size setting (see results in responses to Q2 and Q3).
>
> * Our revision incorporates more rigorous and extensive experiments on different visual tasks: image and video. Following the experimental setting of AdaptFormer except that we use a smaller batch size, our method still shows a good parameter-accuracy trade-off (see results in responses to Q2 with discussion).

---

> > ### Author Response · Authors · 2022-11-19
> > **Response to Reviewer uEYB (Part 2/2)**
> >
> > **Q2 Experiments**
> >
> > **Image downstream tasks**: Thank you for the remark. We had conducted experiments on image datasets but did not incorporate in our first version as the results are not as significant as on the video datasets. Specifically, we use the architectures: ViT-B models from MAE [b] and VideoMAE [c] for three image and two video datasets, respectively. Self-supervised pre-training is used, which follows the main paper settings of AdaptFormer. The results are as follows (also see Table 5 of the revised version), which can draw consistent conclusion in the initial version (i.e., promising parameter-accuracy trade-off):
> >
> > | Method | Avg. Params (M) | CIFAR-100  |  SVHN  |  Food-101  |  SSv2  |  HMDB51  |
> > | :--- | :----: | :----:  | :----:  | :----: | :----:  | :----:  |
> > | Full-tuning | 86.04 (100%) | 85.90 | 97.67 | 90.09 | 53.97 | 46.41 |
> > | Tune FC layer | 0.07 (0.08%) | 69.83 (-16.07) | 66.91 (-30.76) | 69.74 (-20.35) | 29.23 (-24.74) | 49.84 (+3.43) |
> > | VPT | 0.08 (0.09%) | 82.44 (-3.46) | 94.02 (-3.65) | 82.98 (-7.11) | 43.73 (-10.24) | 52.67 (+6.26) |
> > | AdaptFormer-64 | 1.26 (1.46%) | 85.90 (0.00) | 96.89 (-0.78) | 87.61 (-2.48) | 59.02 (+5.05) | 55.69 (+9.28) |
> > | Our ViT-BAPAT-32 | 2.13 (2.47%) | 86.29 (+0.39) | 97.18 (-0.49) | 87.37 (-2.72) | 57.78 (+3.81) | 57.18 (+10.77) |
> > | Our ViT-BAPAT-64 | 3.02 (3.51%) | 86.35 (+0.45) | 97.18 (-0.49) | 87.53 (-2.56) | 57.55 (+3.58) | 57.18 (+10.77) |
> > | Our ViT-BAPAT-128 | 4.79 (5.56%) | 86.47 (+0.57) | 97.28 (-0.39) | 87.66 (-2.43) | 56.97 (+3.00) | 57.70 (+11.29) |
> > | Our ViT-BAPAT-256 | 8.33 (9.68%) | 86.55 (+0.65) | 97.24 (-0.43) | 87.68 (-2.41) | 56.53 (+2.56) | 57.31 (+10.90) |
> >
> > **Discussion**
> >
> > Based on the fairer comparison with AdaptFormer on varied visual tasks, our method still shows promising parameter- accuracy trade-off via much smaller batch size, which is more convenient for reproduction on the general single server with 8 GPUs. We will test our method when a 64-GPU cluster is available for an absolute fair comparison. The underperformance on SSv2 (better than full-tuning) can be due to the smaller batch size as SSv2 is much larger than other compared datasets and can be more relying on larger batch size. In real-world application scenarios, small dataset can be the more common case, which confirms our contribution to the community.
> >
> > **Roughly match the parameters**:
> > We set $d_{bottle}=32$ and do not use adapter, leading to ```1.35M``` parameters for SSv2 and ```1.17M``` for HMDB51 (less than AdaptFormer’s ```1.73M``` and ```1.61M```, respectively). Now, our method in the revised Table 2 shows better performance than AdaptFormer even with less parameters as follows:
> > | Method | $d_{bottle}$  | # Params | SSv2 (Top-1) | # Params | HMDB51 (Top-1) |
> > | :--- | :----: | :----:  | :----:  | :----: | :----:  |
> > | AdaptFormer-Swin | 64 |  1.73M | 40.80% | 1.61M | 68.66% |
> > | Our Swin-BAPAT (w/o Adapter) | 32 | **1.35M** | **46.26%** | **1.17M** | **69.51%** |
> >
> > Regarding compare the performance in multiple scales, our method does not deny the good effect of AdaptFormer but incorporate it into our proposed model design, leading to a model called Swin-BAPAT that shows good parameter-accuracy trade-off on extensive experiments.
> >
> > **Q3 Presentations**
> >
> > Thank you for the correction. We have corrected them in the revision.
> >
> > We added BitFit (Ben-Zaken et al., 2022) in the Section 2.2(Others) and conducted bias tuning for Swin-B. The results are in Table 5 of the revised version as follow:
> >
> > | Method | $d_{bottle}$  | # Params | SSv2 (Top-1) | # Params | HMDB51 (Top-1) |
> > | :--- | :----: | :----:  | :----:  | :----: | :----:  |
> > | Full-tuning | - | 87.82M | 50.99% | 87.69M | 68.07% |
> > | Tune FC Layer | - | 0.18M | 24.13% | 0.05M | 71.28% |
> > | **BitFit** | - | 1.29M | 45.94% | 1.11M | 68.26% |
> > | AdaptFormer-Swin (Adapter) | 64 | 1.73M | 40.80% | 1.61M | 68.66% |
> > | Our Swin-BAPAT (w/o Adapter) | 32 | 1.35M | 46.26% | 1.17M | 69.51% |
> > | Our Swin-BAPAT | 32 | 2.91M | 49.63% | 2.74M | 68.20% |
> >
> > **Discussion**: While without using Adapter, our method still outperforms baselines AdaptFormer-Swin and BitFit with roughly similar number of parameters.
> >
> > ------
> >
> > **References**
> >
> > [a] Chen et al. Exploring Simple Siamese Representation Learning, CVPR 2021
> >
> > [b] He et al. Masked autoencoders are scalable vision learners, CVPR 2022
> >
> > [c] Tong et al. Videomae: Masked autoencoders are data-efficient learners for self-supervised video pre-training, NIPS 2022

---

### Official Review · Reviewer_oYYz · 2022-11-04

**Confidence:** 3
**Correctness:** 3
**Technical Novelty And Significance:** 2
**Empirical Novelty And Significance:** 3
**Recommendation:** 6

**Clarity, Quality, Novelty And Reproducibility:**

This paper clearly explains the proposed method as well as previous relevant baselines such as prefix-tuning and adapter. Since this is more of an empirical paper, the novelty and contribution would mainly come from the quality of the experiments. However, some crucial experiments are missing as described in the Weakness section of the review. The proposed method is not complicated so should not be hard to reproduce. Nevertheless, the authors do not submit code or promise to release code upon acceptance.

**Strength And Weaknesses:**

## Strength

1. This paper proposed a design for parameter efficient fine-tuning and demonstrate its effectiveness in a video setting.

1. The authors conducted some analyses to (partially) validate the design choices of the proposed method.

## Weakness

1. This is mainly an empirical paper. Therefore, the reviewer would expect more extensive and rigorous experiments to demonstrate the generalizability and effectiveness of the proposed method. For example, different architectures, and other combinations of pre-training and fine-tuning datasets.

1. Some conditions are missing in Table 1:
    - Section 2.2 and Figure 2 mentioned three types of most common baselines, but the prompt tuning baseline is missing
    - In CV or NLP, typically more trainable parameters lead to better performance. Therefore, the authors should compare their method with baselines under similar trainable parameters.
    - Since the method is the combination of adapter, PATT (a variant of prefix-tuning), and fc layer fine-tuning, the author should show the results of different combinations of baselines.

1. Some critical ablations of the proposed method are missing:
    - Since the final method is the combination of adapter, PATT, and fc layer fine-tuning, the authors should ablate this.
    - The proposed PATT is a variant of prefix-tuning. Thus the author should also ablate the design of PATT. Specifically, (1) with or without the weight $s$, (2) weighted addition v.s. concatenation of $K_p$ and $V_p$, and (3) $Z^{l-1}$ vs some trainable parameters as the input to Equation 9 for generating $K_p$ and $V_p$.

1. It would also be good to analyze why the proposed design is better (Eg. mathematically, from the perspective of gradient propagation, or anything else). Otherwise, it looks like the authors simply enumerate some possible/common design choices and do a manual architecture search.

**Summary Of The Paper:**

This paper proposed a module inserted into a pre-trained model for parameter efficient fine-tuning. The authors empirically combine adapter, adapt pre-fix tuning, and verify the effectiveness of the design through a series of experiments. The proposed method achieves very competitive results and has the potential to outperform full fine-tuning.

**Summary Of The Review:**

At first glance, the proposed method seems to be a simple yet effective design that combines and adapts previous techniques of parameter efficient fine-tuning. However, after close examination, some critical experiments and ablations are not provided to support this paper.

---

> ### Author Response · Authors · 2022-11-19
> **Response to Reviewer oYYz (Part 1/2)**
>
> Dear Reviewer oYYz,
>
> Thank you for your valuable comments. We promise here to release our code upon acceptance and will follow up the coming reproduction issues on GitHub for sure. We conducted some crucial experiments and have included all discussions and results in our revised manuscript (blue content in the main text).
>
> **Q1 More extensive and rigorous experiments.**
> Thank you for the remark. We had conducted experiments on image datasets but did not incorporate in our first version as the results are not as significant as on the video datasets. Specifically, we use the architectures: ViT-B models from MAE [a] and VideoMAE [b] for three image and two video datasets, respectively. Self-supervised pre-training is used, which follows the main paper settings of AdaptFormer. The results are as follows (also see Table 5 of the revised version), which can draw consistent conclusion in the initial version (i.e., promising parameter-accuracy trade-off):
>
> | Method | Avg. Params (M) | CIFAR-100  |  SVHN  |  Food-101  |  SSv2  |  HMDB51  |
> | :--- | :----: | :----:  | :----:  | :----: | :----:  | :----:  |
> | Full-tuning | 86.04 (100%) | 85.90 | 97.67 | 90.09 | 53.97 | 46.41 |
> | Tune FC layer | 0.07 (0.08%) | 69.83 (-16.07) | 66.91 (-30.76) | 69.74 (-20.35) | 29.23 (-24.74) | 49.84 (+3.43) |
> | VPT | 0.08 (0.09%) | 82.44 (-3.46) | 94.02 (-3.65) | 82.98 (-7.11) | 43.73 (-10.24) | 52.67 (+6.26) |
> | AdaptFormer-64 | 1.26 (1.46%) | 85.90 (0.00) | 96.89 (-0.78) | 87.61 (-2.48) | 59.02 (+5.05) | 55.69 (+9.28) |
> | Our ViT-BAPAT-32 | 2.13 (2.47%) | 86.29 (+0.39) | 97.18 (-0.49) | 87.37 (-2.72) | 57.78 (+3.81) | 57.18 (+10.77) |
> | Our ViT-BAPAT-64 | 3.02 (3.51%) | 86.35 (+0.45) | 97.18 (-0.49) | 87.53 (-2.56) | 57.55 (+3.58) | 57.18 (+10.77) |
> | Our ViT-BAPAT-128 | 4.79 (5.56%) | 86.47 (+0.57) | 97.28 (-0.39) | 87.66 (-2.43) | 56.97 (+3.00) | 57.70 (+11.29) |
> | Our ViT-BAPAT-256 | 8.33 (9.68%) | 86.55 (+0.65) | 97.24 (-0.43) | 87.68 (-2.41) | 56.53 (+2.56) | 57.31 (+10.90) |
>
> **Discussion**
>
> Based on the extended comparison with AdaptFormer on varied visual tasks, our method still shows promising parameter- accuracy trade-off via much smaller batch size, which is more convenient for reproduction on the general single server with 8 GPUs. We will test our method when a 64-GPU cluster is available for an absolute fair comparison. The underperformance on SSv2 (better than full-tuning) can be due to the smaller batch size as SSv2 is much larger than other compared datasets and can be more relying on larger batch size. In real-world application scenarios, small dataset can be the more common case, which confirms our contribution to the community.
>
> **Q2 Some missing conditions.**
>
> **Q2.1 Prompt tuning baseline is missing**:
> We compare the VPT using the settings of AdaptFormer. Please refer to the response to Q1.
>
> **Q2.2 Under similar trainable parameters**:
> We set $d_{bottle}=32$ and do not use adapter, leading to ```1.35M``` parameters for SSv2 and ```1.17M``` for HMDB51 (less than AdaptFormer’s ```1.73M``` and ```1.61M```, respectively). Now, our method in the revised Table 2 shows better performance than AdaptFormer even with less parameters as follows:
> | Method | $d_{bottle}$  | # Params | SSv2 (Top-1) | # Params | HMDB51 (Top-1) |
> | :--- | :----: | :----:  | :----:  | :----: | :----:  |
> | AdaptFormer-Swin | 64 |  1.73M | 40.80% | 1.61M | 68.66% |
> | Our Swin-BAPAT (w/o Adapter) | 32 | **1.35M** | **46.26%** | **1.17M** | **69.51%** |
>
> **Q2.3 Results of different combinations of adapter, PATT, and FC layer**:
> Previous version mainly lacks the ablation of Adapter. Hence, we extend the ablation of without using Adapter as follows (also see Table 5 of the revised version):
> | Method | $d_{bottle}$  | # Params | SSv2 (Top-1) | # Params | HMDB51 (Top-1) |
> | :--- | :----: | :----:  | :----:  | :----: | :----:  |
> | Full-tuning | - | 87.82M | 50.99% | 87.69M | 68.07% |
> | Tune FC Layer | - | 0.18M | 24.13% | 0.05M | 71.28% |
> | AdaptFormer-Swin (Adapter) | 64 | 1.73M | 40.80% | 1.61M | 68.66% |
> | Our Swin-BAPAT (w/o Adapter) | 32 | 1.35M | 46.26% | 1.17M | 69.51% |
> | Our Swin-BAPAT (w/o Adapter) | 64 | 2.51M | 49.23% | 2.34M | 71.34% |
> | Our Swin-BAPAT (w/o Adapter) | 128 | 4.83M | 52.57% | 4.65M | 70.56% |
> | Our Swin-BAPAT (w/o Adapter) | 256 | 9.45M | 52.71% | 9.27M | 70.23% |
> | Our Swin-BAPAT | 32 | 2.91M | 49.63% | 2.74M | 68.20% |
> | Our Swin-BAPAT | 64 | 4.07M | 51.80% | 3.89M | 70.10% |
> | Our Swin-BAPAT | 128 | 6.38M | 53.36% | 6.20M | 71.93% |
> | Our Swin-BAPAT | 256 | 11.00M | 53.98% | 10.83M | 69.64% |
>
> Please note that FC layer is commonly used for all the compared baselines on SSv2. We discussed this for small scale dataset HMDB51 in the supplementary.

---

> > ### Author Response · Authors · 2022-11-19
> > **Response to Reviewer oYYz (Part 2/2)**
> >
> > **Q3 Critical ablations.**
> >
> > **Q3.1 Ablate the combinations of adapter, PATT, and fc layer fine-tuning**:
> > Thank you for the suggestion. Please refer to the response to Q2.3.
> >
> > **Q3.1 Ablate the design of PATT**:
> > When we set weight ``` s=1.0```, it means without taking the weight. Table 3 shows this ablation. For the “(2) weighted addition v.s. concatenation of Kp and Vp, and (3) Zl−1 vs some trainable parameters as the input to Equation 9 for generating Kp and Vp.”, we conducted experiments and incorporate this further ablation in Table 4 as follows.
> > | Method  | # Params | SSv2 (Top-1) | # Params | HMDB51 (Top-1) |
> > | :---  | :----:  | :----:  | :----: | :----:  |
> > | Concat (**K**, **V**) |  6.38M | 15.61% | 6.20M | 20.98% |
> > | No $Z^{l-1}$ (**K**, **V**) | 8.74M | 51.06% | 8.56M | 67.41% |
> > | Ours (**K**, **V**) | 6.38M | 53.38% | 6.20M | 71.41% |
> >
> > **Q4 Analyze why the proposed design is better.**
> >
> > Thank you for this valuable question. We have been thinking about this question since working on this direction. We had investigated most of relevant works in NLP and CV domains but did not find strong mathematical proof regarding why existing parameter-efficient baselines work. The current empirical finding in NLP [c] is that when the model’s parameter scale is over billion level, existing method such prompt-tuning can match full-tuning, which is not hold in CV domain as existing visual models do not reach that level. Most existing parameter-efficient works are inspired by the NLP domain while there remains lack of theoretical explanation regarding why these models work even in the NLP domain.
> > We attempted to answer this question in Section 2.4 by fine-tuning individual modules of the Video Swin Transformer. Findings show that the QKV and MLP modules have more parameters than other modules and can lead to better fine-tuning performance. Hence, tuning which position can be important for the parameter-efficient transfer learning. This is the first work that analysis prefix-tuning for visual tasks at the attention module as existing works usually do it at MLP module (i.e., Adapter) or inter-transformer layers (Prompt Tuning). Based on extensive experiments, both Adapter and our PATT are good for parameter-efficient transfer learning, leading to our proposed model Swin-BAPAT. We hope that our extended empirical analysis can justify our contribution and we will continue to work on the direction of finding theoretical support in the mathematic and optimization perspectives.
> >
> > ------
> >
> > **References**
> >
> > [a] He et al. Masked autoencoders are scalable vision learners, CVPR 2022
> >
> > [b] Tong et al. Videomae: Masked autoencoders are data-efficient learners for self-supervised video pre-training, NIPS 2022
> >
> > [c] Lester et al. The Power of Scale for Parameter-Efficient Prompt Tuning, EMNLP 2021

---

> ### Comment · Reviewer_oYYz · 2022-11-20
> **Thanks for your efforts on the rebuttal**
>
> Thanks for your efforts! The reviewer has gone through the rebuttal. Most of the concerns about the lack of critical experiments and ablations have been properly addressed in the rebuttal. Based on current results, the reviewer tends to believe this is an effective design for parameter efficient training. Overall, the reviewer decided to raise the score to 6.
>
> It would still be good to add some more experiments such as using different pre-trained models and other model architectures, as well as provide some theoretical analysis to make this paper stronger. Nevertheless, the reviewer agrees that theoretical analysis of the current design is a difficult research problem, and acknowledges that the time for rebuttal is limited for large-scale analyses.

---

> > ### Author Response · Authors · 2022-12-09
> > **Thank You and Our New Results.**
> >
> > Dear Reviewer oYYz
> >
> > Thank you very much for raising the score and for the suggestions.
> >
> > We conduct further experiments for image and video tasks using weights of supervised pre-training. For image tasks, we use the pre-trained weights on ImageNet_21K [vit_base_patch16_224_miil_in21k](https://miil-public-eu.oss-eu-central-1.aliyuncs.com/model-zoo/ImageNet_21K_P/models/timm/vit_base_patch16_224_in21k_miil.pth). For video tasks, we use ViT-B of VideoMAE [fine-tuned on Kinetics-400 (Top-1=81.5)]( https://drive.google.com/file/d/1MzwteHH-1yuMnFb8vRBQDvngV1Zl-d3z/view?usp=sharing). Since [AdaptFormer](https://github.com/ShoufaChen/AdaptFormer/releases) has not provided the supervised pre-trained weights, our supervised pre-training settings might differ from the ones used by AdaptFormer. The results are as follows:
> >
> > **Table 1**:  Comparison of Top-1 accuracy via ViT-B models with supervised pre-training for image and video tasks. * indicates reproduction via our pre-training weights.
> > | Method | Avg. Params (M) | CIFAR-100  |  SVHN  |  Food-101  |  SSv2  |  HMDB51  |
> > | :--- | :----: | :----:  | :----:  | :----: | :----:  | :----:  |
> > | Full-tuning* | 86.04 (100%) | 90.00 | 97.54 | 89.70 | 51.36 | 57.51 |
> > | Tune FC layer* | 0.07 (0.08%) | 60.44 (-29.56) | 60.44 (-37.10) | 54.07 (-35.63) | 37.52 (-13.84) | 68.07 (+10.56) |
> > | AdaptFormer-64* | 1.26 (1.46%) | 90.80 (+0.80) | 96.73 (-0.81) | 89.15 (-0.55) | 54.70 (+3.34) | 66.03 (+8.52) |
> > | Our ViT-BAPAT-32 | 2.13 (2.47%) | 91.79 (+1.79) | 96.97 (-0.57) | 89.43 (-0.27) | 59.49 (+8.13) | 73.31 (+15.80) |
> > | Our ViT-BAPAT-64 | 3.02 (3.51%) | 91.55 (+1.55) | 96.91 (-0.63) | 89.63 (-0.07) | 59.32 (+7.96) | 73.38 (+15.87) |
> > | Our ViT-BAPAT-128 | 4.79 (5.56%) | 91.45 (+1.45) | 97.05 (-0.49) | 89.48 (-0.22) | 58.70 (+7.34) | 73.83 (+16.32) |
> > | Our ViT-BAPAT-256 | 8.33 (9.68%) | 91.57 (+1.57) | 96.88 (-0.66) | 89.43 (-0.27) | 57.58 (+6.22) | 73.97 (+16.46) |
> >
> > We hope the above further experiments can make the effectiveness of our model design more convincing. Thank you again for your valuable time and suggestions that help us improve the paper.
> >
> > Best regards,
> >
> > Authors

---

### Official Review · Reviewer_GJuJ · 2022-11-04

**Confidence:** 4
**Clarity, Quality, Novelty And Reproducibility:**
**Correctness:** 3
**Technical Novelty And Significance:** 3
**Empirical Novelty And Significance:** 3
**Recommendation:** 6

**Details Of Ethics Concerns:**



**Strength And Weaknesses:**

Strengths:
● The paper is well written and well structured.
● The authors considers the situation of insufficient computing resources. Under the corresponding experimental settings, the experimental results demonstrate the superiority of the method proposed in paper.
Weakness:
● As one of the baselines, Adaptformer can achieve excellent performance (59.02 on SSv2 and 55.69 on HMDB51) after larger batch size training.  Why does reducing the batch size has such a big impact on performance of Adaptformer? It is uncertain whether authors have explored the hyperparameters of Adaptformer under small batch size setting. In other words, authors can increase the batch size to compare the proposed method and Adaptformer.
● The paper only shows the experimental results but does not elaborate on the reasons for the gains brought by the method.
Minors:
● There are some minor mistakes in the experimental table: first two lines in Table 1 appear to have numerical errors where "0.11M" is inconsistent with"0.18M" in Table 3, and "Tune FC layer" and "Full-tuning" should be swapped.

**Summary Of The Paper:**

This paper proposes a new framework to adapt large video-based models to down-stream tasks with a parameter-accuracy trade-off. It analyzes different PETL techniques and investigates the importance of fine-tuning position of their methods. In order to better transfer prefix-tuning from NLP to vision task, it compares differences between NLP and video data regarding data structures and pre-training mechanisms.

**Summary Of The Review:**

The proposed method seems to achieve the best performance under the condition of limited computing resources. This paper should demonstrate the superiority of the method under fairer conditions or give sufficient reason analysis.

---

> ### Author Response · Authors · 2022-11-19
> **Response to Reviewer GJuJ (Part 1/2)**
>
> Dear Reviewer GJuJ,
>
> Thank you for your valuable comments. We demonstrate the superiority of the method under fairer conditions. We have included all discussions and results in our revised manuscript (blue content in the main text).
>
> **Q1 Big impact of reducing the batch size.**
>
> This is a common issue on video-based tasks and empirically verified reality on different datasets especially large-scale ones such as SSv2. This can also be observed from ST-Adapter mentioned by Reviewer EH6p where full-tuning the ViT with ImageNet-21k can perform 40% via 128 batch size on SSv2 while 41.5% via 1,024 batch size. Another work explored the effect of batch size on visual tasks is SimSiam [a], which uses consistent hyperparameter for all experiments and shows that batch size 512 and 1,024 can gain around 2% improvement over the batch size of 64 on the ImageNet linear evaluation. We had explored the hyperparameter of initial learning rate (e.g., 0.1, 0.01, and 0.001) and 0.1 turned out to work better. Based on this exploration on hyperparameters and relevant works investigating batch size [a], we followed the original settings of AdaptFormer except the smaller batch size.
>
> The mentioned excellent performance of AdaptFormer (59.02% on SSv2 and 55.69% on HMDB51) are based on the ViT-16 backbone with self-supervised pre-training, which is different with our experimental setting regarding the backbone and pre-trained weights. Please note that AdaptFormer’s result on HMDB51 is not promising under the setting of self-supervised pre-training. Our result on SSv2, based on Swin-B pre-trained on Kinetics 600, is 54.06%, outperforming full-tuning with a larger margin (3.07%, from 50.99% to 54.06%) than the corresponding result of AdapFormer using large training batch-size (1.17%, from 52.92% to 54.09%) by using slightly more parameters. The corresponding results on AdaptFormer using large batch-size (i.e., 1024) are (54.09 on SSv2 and 74.65 on HMDB51).
>
> **Extensive experiments under fairer conditions**: We had conducted experiments on image datasets but did not incorporate in our first version as the results are not as significant as on the video datasets. Specifically, we use the architectures: ViT-B models from MAE [b] and VideoMAE [c] for three image and two video datasets, respectively. Self-supervised pre-training is used, which follows the main paper settings of AdaptFormer. The results are as follows (also see Table 5 of the revised version), which can draw consistent conclusion in the initial version (i.e., promising parameter-accuracy trade-off):
>
> | Method | Avg. Params (M) | CIFAR-100  |  SVHN  |  Food-101  |  SSv2  |  HMDB51  |
> | :--- | :----: | :----:  | :----:  | :----: | :----:  | :----:  |
> | Full-tuning | 86.04 (100%) | 85.90 | 97.67 | 90.09 | 53.97 | 46.41 |
> | Tune FC layer | 0.07 (0.08%) | 69.83 (-16.07) | 66.91 (-30.76) | 69.74 (-20.35) | 29.23 (-24.74) | 49.84 (+3.43) |
> | VPT | 0.08 (0.09%) | 82.44 (-3.46) | 94.02 (-3.65) | 82.98 (-7.11) | 43.73 (-10.24) | 52.67 (+6.26) |
> | AdaptFormer-64 | 1.26 (1.46%) | 85.90 (0.00) | 96.89 (-0.78) | 87.61 (-2.48) | 59.02 (+5.05) | 55.69 (+9.28) |
> | Our ViT-BAPAT-32 | 2.13 (2.47%) | 86.29 (+0.39) | 97.18 (-0.49) | 87.37 (-2.72) | 57.78 (+3.81) | 57.18 (+10.77) |
> | Our ViT-BAPAT-64 | 3.02 (3.51%) | 86.35 (+0.45) | 97.18 (-0.49) | 87.53 (-2.56) | 57.55 (+3.58) | 57.18 (+10.77) |
> | Our ViT-BAPAT-128 | 4.79 (5.56%) | 86.47 (+0.57) | 97.28 (-0.39) | 87.66 (-2.43) | 56.97 (+3.00) | 57.70 (+11.29) |
> | Our ViT-BAPAT-256 | 8.33 (9.68%) | 86.55 (+0.65) | 97.24 (-0.43) | 87.68 (-2.41) | 56.53 (+2.56) | 57.31 (+10.90) |
>
> **Discussion**
>
> Based on the fairer comparison with AdaptFormer on varied visual tasks, our method still shows promising parameter- accuracy trade-off via much smaller batch size, which is more convenient for reproduction on the general single server with 8 GPUs. We will test our method when a 64-GPU cluster is available for an absolute fair comparison. The underperformance on SSv2 (better than full-tuning) can be due to the smaller batch size as SSv2 is much larger than other compared datasets and can be more relying on larger batch size. In real-world application scenarios, small dataset can be the more common case, which confirms our contribution to the community.

---

> > ### Author Response · Authors · 2022-11-19
> > **Response to Reviewer GJuJ (Part 2/2)**
> >
> > **Q2 Recap reasons for the gains brought by the method.**
> >
> > The current empirical finding in NLP [d] is that when the model’s parameter scale is over billion level, existing method such prompt-tuning can match full-tuning, which is not hold in CV domain as existing visual models do not reach that level. Most existing parameter-efficient works are inspired by the NLP domain while there remains lack of theoretical explanation regarding why these models work even in the NLP domain. We have been thinking about this question and attempted to answer this question in Section 2.4 by fine-tuning individual modules of the Video Swin Transformer. Findings show that the QKV and MLP modules have more parameters than other modules and can lead to better fine-tuning performance. Hence, tuning which position can be important for the parameter-efficient transfer learning. This is the first work that analysis prefix-tuning for visual tasks at the attention module as existing works usually do it at MLP module (i.e., Adapter) or inter-transformer layers (Prompt Tuning). Based on extensive experiments, both Adapter and our PATT are good for parameter-efficient transfer learning, leading to our proposed model Swin-BAPAT. We hope that our empirical analysis in Section 2.4 and the extended fairer comparison can justify the superiority of the method and we will continue to work on the direction of finding theoretical support in the mathematic and optimization perspectives.
> >
> > **Q3 Minor mistakes in the experimental table.**
> >
> > Thank you very much for pointing out the mistake in Table 1. We have corrected it.
> >
> > ------
> >
> > **References**
> >
> > [a] Chen et al. Exploring Simple Siamese Representation Learning, CVPR 2021
> >
> > [b] He et al. Masked autoencoders are scalable vision learners, CVPR 2022
> >
> > [c] Tong et al. Videomae: Masked autoencoders are data-efficient learners for self-supervised video pre-training, NIPS 2022
> >
> > [d] Lester et al. The Power of Scale for Parameter-Efficient Prompt Tuning, EMNLP 2021

---

### Author Response · Authors · 2022-11-19
**Summary of our rebuttal and discussion**

Dear Reviewers and ACs:

We really appreciate your valuable time and efforts in reviewing our paper. We are encouraged to find reviewers’ recognition of our contribution as follows:

* **Important topic and well-motivated work**: The motivation for applying parameter-efficient learning to videos is sound and reasonable. [GJuJ,uEYB] First work that leverages the prefix-tuning to visual tasks and made modifications based on a comprehensive understanding of the structural difference between NLP and vision data. [oYYz, EH6p]

* **Effective and efficient method in Video setting**: Ablation studies and empirical results are strong and convincing. [uEYB] Under the corresponding experimental settings, the experimental results demonstrate the superiority of the method proposed in the paper. [GJuJ] The proposed method achieves very competitive results and has the potential to outperform full fine-tuning. [oYYz] Several analyses has been presented on the hyper-parameters (Tables 2, 3, 4, and 5). [EH6p]

* **Well written paper and clearly-presented method**: The paper is well written, well structured and easy to understand. [GJuJ, oYYz, uEYB, EH6p]

We are thankful for reviews insightful and constructive suggestions that help us improve the paper. In the following, we summarize the major revisions incorporated in our rebuttal version.
* **Direct Comparison with AdaptFormer on more visual tasks**: We add extensive experiments on three image datasets and two video datasets by following the setting in AdaptFormer’s main paper (see Table 5). [GJuJ, oYYz, uEYB, EH6p]

* **More ablation on the model design (i.e., FC layer, Adapter, and PATT)**: We conduct more ablations and add them to Table 2 [GJuJ, oYYz, uEYB, EH6p] and Table 4 [oYYz].
* **Source Code**: We will release the code on GitHub for sure.

We would like to thank all reviewers and ACs again.

Best Regards,

Authors

---

### Author Response · Authors · 2022-12-09
**Sincerely Look Forward to Your Feedback**

Dear Reviewers and ACs:

Thanks again for your valuable time and constructive suggestions, helping us improve the completeness of this paper. We hope that our added experiments and ablations can convince you more the merits of our work. **We will release all the pre-training weights (both of the main paper and of new results during rebuttal) on GitHub for reproduction**.

Since the deadline for discussion is approaching, please feel free to let us know if there is any additional clarification or experiment that we can provide. We appreciate your further suggestions.

Best regards,

Authors

---

### Decision · Program_Chairs · 2023-01-20

**Decision:**

Reject

**Justification For Why Not Higher Score:**

There are a number of concerns about exactly the empirical contributions of the paper, either targeted towards videos or shown more generally for vision tasks, and hence further work is needed to tighten the approach and experimental methodology.

**Justification For Why Not Lower Score:**

N/A

**Metareview: Summary, Strengths And Weaknesses:**


This paper looks at the problem of parameter-efficient transfer learning for computer vision. Inspired by prior methods, the authors combine several known principles including adapters and pre-fix tuning for self-attention and MLP layers. Results are demonstrated across video vision tasks, including with self-supervised pretrained models, demonstrating improved parameter-efficiency and in some cases improvements over full finetuning.

Overall, the reviewers appreciated the writing of the paper, the importance of the topic, and the results demonstrated (though caveats forthcoming). However, a range of weaknesses were identified, including: 1) The comparisons to baselines such as Adaptformer and VPT, including hyperparameter settings and fair comparisons (uEYB), 2) The fact that the method itself is not intrinsic to videos, yet did not present results on image datasets (uEYB,EH6p), 3) Explanations or analysis for why the method can work well (GJuJ, oYYz), and 4) Lack of extensive and rigorous experimentation to demonstrate the effectiveness and generality of the approach, including missing conditions and experiments/ablations (oYYz,EH6p), 4) Novelty with respect to prior findings in these areas (uEYB,EH6p). The authors provided extensive rebuttals, including new experiments on image-based datasets.

However, after extensive discussions there are still considerable concerns that have not been addressed. Ultimately, this is an empirical paper with a simple method that combines ideas from prior works and demonstrates results empirically. Ignoring incremental novelty and the concurrent work, such empirical methods still need strong motivation/descriptions of where and why the method works, rigorous empirical methodology and experimental results (including rigorous implementations of baselines with equal hyper-parameter tuning budgets given to all methods), and demonstration that the method is general across many application areas (or at least description of why it's not). In its current state, the paper does not live up to this however. Specifically, some examples of remaining concerns include: 1) Unclear motivation: It's not clear how the method benefits video tasks in particular (which was the focus of the paper), 2) The image-based experiments (and indeed the video-based ones as well, based on the initial reviews) highlight a number of concerns including potentially weak implementations of baselines (e.g. using VPT-shallow and/or not matching the number of parameters across these baselines and the proposed method), and 3) Still lacks extensive experiments across many architectures, downstream tasks, etc. Either the method should be well-motivated to the limited tasks (videos) or more extensive experiments should be done to show the generality of the method.

Given these significant concerns highlighted during the reviews and ensuing discussion, and the fact that these points hit directly the experiments that are crucial for an empirical study, I cannot recommend acceptance in its current state.